# In Silico Reconstruction of Sperm Chemotaxis

**DOI:** 10.3390/ijms22179104

**Published:** 2021-08-24

**Authors:** Masahiro Naruse, Midori Matsumoto

**Affiliations:** Department of Bioscience and Informatics, Keio University, Yokohama 223-8522, Japan; masahiro_naruse@yahoo.co.jp

**Keywords:** fertilization, sperm chemotaxis, modeling of cell, desensitization

## Abstract

In echinoderms, sperm swims in random circles and turns in response to a chemoattractant. The chemoattractant evokes transient Ca^2+^ influx in the sperm flagellum and induces turning behavior. Recently, the molecular mechanisms and biophysical properties of this sperm response have been clarified. Based on these experimental findings, in this study, we reconstructed a sperm model in silico to demonstrate an algorithm for sperm chemotaxis. We also focused on the importance of desensitizing the chemoattractant receptor in long-range chemotaxis because sperm approach distantly located eggs, and they must sense the chemoattractant concentration over a broad range. Using parameters of the sea urchin, simulations showed that a number of sperm could reach the egg from millimeter-order distances with desensitization, indicating that we could organize a functional sperm model, and that desensitization of the receptor is essential for sperm chemotaxis. Then, we compared the model with starfish sperm, which has a different desensitization scheme and analyzed the properties of the model against various disturbances. Our approach can be applied as a novel tool in chemotaxis research.

## 1. Introduction

During fertilization, sperm must reach the oocyte from far away, and it is guided by the diffused chemoattractant dispersed from the egg or egg coat in many animals. Unique chemoattractants have been identified from different species, including resact in the sea urchin *Arbacia punctulata* [1], a group of unsaturated fatty acids in the coral *Montipora digitata* [2], asterosap (Asap) in the starfish *Asterias amurensis* [3], Allurin in the amphibian *Xenopus laevis* [4], sperm-activating and -attracting factor (SAAF) in the ascidian *Ciona intestinalis* [5], and progesterone in mammals [6,7] Additionally, chemoattractant receptors on the sperm surface have been identified in some species, such as the CatSper channel for progesterone in *Homo sapiens* [8,9,10], guanylyl cyclase (GC) for Asap in *A. amurensis* [11], and resact in *Ar. punctulata* [12]. These chemoattractants and their receptors are the key molecules for sperm attraction. Intracellular signaling controls sperm trajectory and is a complex process that varies by species; however, [Ca^2+^]_i_ is closely tied to the sperm flagella-beating pattern, invariant across species [13,14].

In echinoderms, masses of sperm and eggs are released in the sea for external fertilization, and sperm go upstream along the chemoattractant gradient toward the egg. In this process, each sperm swims in a circle without stimulation, and the transient increase in [Ca^2+^]_i_ evoked by chemoattractant stimulation induces their turning behavior [15]. The sequence of turns results in a chemotactic movement toward the egg. In *Ar. punctulata* and *A. amurensis*, the chemoattractant receptor GC produces cGMP in response to chemoattractant stimulation, and an increase in [cGMP]_i_ induces a K^+^ efflux through the cyclic nucleotide-gated channel (CNGK), which is followed by a subsequent increase in pH_i_, and finally, a Ca^2+^ influx is induced [16,17,18,19,20]. In this study, we used a system biological approach, i.e., we reconstructed sperm chemotaxis *in silico* based on the results of previous research to confirm whether the sperm model would show chemotactic behavior in the simulation.

Regarding chemotaxis of *Ar. punctulata* sperm, the biophysical properties of GC, cGMP, and CNGK have been precisely investigated, and the contribution of GC-desensitization in the broad-range sensing of resact has been suggested [21]. Therefore, we modeled its sperm, and we used the starfish *A. amurensis*, which has been used as a fertilization model in our laboratory [22,23]. Although the GCs of both sea urchin and starfish sperm are inactivated by dephosphorylation [24,25], the desensitization/dephosphorylation reactions are different. In sea urchins, GC, which binds to resact, can undergo autodephosphorylation [21]. However, GC dephosphorylation requires PP2A activity in starfish [25], similar to the natriuretic peptide receptor (NPR), the human GC homolog, which is dephosphorylated by protein kinase C (PKC) and the PP2A-mediated pathway [26,27,28,29]. Therefore, we assumed that the GC of starfish sperm is desensitized/dephosphorylated when Ca^2+^ influx occurs as in the case of NPR, meaning that GC is dephosphorylated by the initiation of each turning behavior. Thus, in this study, we compared continuous desensitization in sea urchins and discrete desensitization in starfish using sperm models (Figure 1A).

Mathematical models have been utilized to organize macroscopic phenomena from chaotic elements at the microscopic level and provide new insights for research [30,31]. In this study, we developed functional models of sperm chemotaxis, a macroscopic cellular movement, using molecular and cellular experimental data. Our approach in this study is expected to provide new insights into the study of sperm chemotaxis.

## 2. Results

### 2.1. Modeling of Sperm

The sperm model was composed of a circular-motion mode and a turn mode (Figure 1A). Each sperm swims in a circle and then turns stochastically, depending on the local concentration of chemoattractant ((Chemoattractant)) and the amount of active/phosphorylated GC ((GC)) on the sperm surface. Sperm trajectories of these two modes and unit time (60 ms) were defined based on a previous investigation [15]. Then, the changes in the curvature of both sea urchin sperm and starfish sperm were approximated by linear or quadratic functions (Figure 1B,D), and these curvature patterns drew trajectories nearly identical to those in the previous report by Böhmer et al. [15] (Figure 1C,E). The transition from the circular mode to the turn mode occured if the amount of intracellular cGMP ([cGMP]i) exceeded the threshold, and a refractory period existed after the turn mode. Because the CNGK channel required several dozen cGMP molecules to respond to *Ar. punctulate* sperm [18], we set the threshold for [cGMP]_i_ to 100 for both sea urchin and starfish models.

Although all parameters could be determined for the sea urchin model using the results of biophysical measurements from previous studies, the cGMP generation/threshold and GC-Asap binding in starfish have never been precisely measured. Accordingly, we assumed that the parameters of starfish were similar to those of sea urchins because the K_D_ value of GC-Asap binding in starfish is 57 pM and GC-resact binding in sea urchins is 90 pM [11,16]. The response against cGMP exposure in starfish sperm has been reported to be almost identical to that in sea urchin sperm [11,15]. Therefore, the parameters of the starfish model were supplemented by sea urchin parameters.

GC-chemoattractant binding was constructed based on a previous report [21], in which binding between GC and the chemoattractant had negative cooperativity with K_1/2_ = 0.65 ± 0.08 nM and a Hill coefficient, h = 0.49 ± 0.03. Thus, the amount of binding was calculated using (GC) × ([Chemoattractant]^h^)/(K_1/2_^h^ + [Chemoattractant]^h^)/2000, where division by 2000 resulted from the binding velocity because the binding process between GC and chemoattractants took approximately 2 min to achieve equilibrium [11]. Additionally, GC function was modeled as a single GC producing 4.3 ± 1.7 cGMP per unit time with chemoattractant binding, and the lifetime of cGMP was limited in that unit time. The parameters with SD were given by a normal distribution each time. 

In sea urchin sperm, GC, which binds to resact, was desensitized in approximately 160 ms. Therefore, the amount of chemoattractant-binding GC generated at a certain time was maintained for three units and then subtracted from [GC] in the model. However, desensitization of starfish sperm occurred only with the initiation of turning behavior. Therefore, the amount of GC that bound with Asap when the transition to the turn mode occurs was subtracted from [GC] in the model. The initial [GC] of the sea urchin and starfish models were 300,000 and 110,000, respectively, based on experimental measurements [11,21].

The distribution of chemoattractant was calculated using the time-independent 2D diffusion equation ∂^2^C/∂r^2^ + 1/r × (∂C/∂r) = 0 under the boundary condition of no chemoattractant at the edge of the defined area. The egg radius was set to 100 μm, including the egg coat. Figure 1F showes the shape of the chemoattractant gradient when the field was defined as a 5000 μm diameter circle area, and the chemoattractant concentration at the egg surface ((Chemoattractant)_s_) was 100 pM.

### 2.2. Performance of Chemotaxis Model in Sea Urchin

The above-described model was implemented using C language (see Appendix A). Figure 2A,B show the results of simulations with or without desensitization, where the defined area was the same as that in Figure 1F, and the length of the simulation was 30 min. The distance between the sperm start-point and the egg center was varied from 200 to 5000 μm. As shown in Figure 2A, approximately 1.5% of the number of sperm reached the egg from as far as 2000 μm, whereas simulations without desensitization produced much lower arrival ratios. Eight representative examples, four successes, and four failures were shown in Figure 2C. Trajectories 2, 4, 6, and 8 showed successful examples; trajectories 1, 3, 5, and 7 represent failures. Because the rotational direction of sperm was set in a clockwise fashion in this simulation, the sperm trajectory heading toward the egg also tended to slightly revolve around a center point. Although we used a 5000 μm diameter field in these simulations, the 2500 or 7500 μm field showed similar results (Appendix A).

### 2.3. Performance of Chemotaxis Model in Starfish

Similar to the sea urchin, the starfish model was implemented using C language (see Appendix A). Figure 3A,B show the results of the simulations, where the conditions of the simulations are the same as those of the sea urchin (Figure 2A,B). These results showed that more than 2% of the number of sperm reached the egg from 2000 μm. Conversely, simulations without desensitization produced a much lower arrival ratio at distances greater than 200 μm, whereas virtually no sperm reached the egg from distances greater than 300 μm. The arrival ratio of the starfish model decreased faster than that of the sea urchin model with increasing distance in short-range chemotaxis (less than 1200 μm distance), whereas the starfish ratio exceeded that of the sea urchins in the 1300 μm or longer range, and some populations could even reach the egg from a 4.6 mm distance, close to the edge of the defined field. Representative trajectories are shown in Figure 3C.

### 2.4. Further Analysis of Models

Next, the robustness of these sperm models against the modification in parameters, the threshold for [cGMP]_i_, K_1/2_, and [Chemoattractant]_s_, was examined. Figure 4A–F show the changes in the arrival rate to the egg with such parameter modifications for two different tasks (500 and 1000 μm distance with a time limit of 30 min). Although sperm models of both sea urchin and starfish were robust, the results of the starfish model (Figure 4B,D,F) showed greater robustness than that of the sea urchins (Figure 4A,C,E). The two tasks showed similar tendencies in the sea urchin model. However, in the starfish model, the 500 μm task with modifications of K_1/2_ or [Chemoattractant]_s_ resulted in major differences from the others but retained high arrival rates to the egg, suggesting that starfish are tolerant to short-distance chemotaxis.

To examine the effect of physical disturbance, fluctuations in the relative position between the egg and sperm were fed into the simulation. Here, we choose a sine-wave pattern oscillation along the *y*-axis, where sperm started from (500, 0) or (1000, 0). The amplitude was fixed at 500 μm, and the frequency was varied from 10^–5^ to 10^5^ Hz. As shown in Figure 4G,H, the two different tasks had similar tendencies in each, and the results indicated that the model showed broad robustness for such fluctuations. Interestingly, a higher sperm-arrival rate was observed at 0.01 Hz (broken line in Figure 4G,H) or higher frequency than the no-fluctuation condition. Only the sea urchin model showed a local maximum at approximately 100 Hz (arrow in Figure 4G), whereas the starfish model showed a stable increase at 0.01 Hz or higher frequency vibration.

## 3. Discussion

Our concept of desensitization theory for long-range chemotaxis was shown in the Graphic Abstract. The driving force to shift sperm toward the egg came from the difference in turn occurrence probabilities in the swimming circle. In Figure 1F, (Chemoattractant) at 4950 μm, 3500 μm, and 500 μm distances from the egg were 257 fM, 9 pM, and 59 pM, respectively. Thus, they were highly diverse. However, for example, if sperm were desensitized 0% at 4950 μm, 80.8% at 3500 μm, and 91.1% at 500 μm, respectively, the amount of binding GC in the sea urchin model was approximately 6300, regardless of distance. Thus, desensitization could tune sperm reactivity to a higher chemoattractant concentration, similar to olfactory adaptation [32]. In theory, excess desensitization will make sperm too insensitive. Conversely, insufficiently desensitized sperm could not find the correct direction toward the egg because it would be stimulated anywhere in the swimming circle. Although the detailed mechanism of dephosphorylation in starfish has not been determined, the involvement of PKC and PP2A has been suggested in GC dephosphorylation, and results indicated that long-range chemotaxis is disturbed by excess or insufficient desensitization (Appendix A).

The results showed that simulations without desensitization lost the chemotactic function (Figure 2A,B and Figure 3A,B), suggesting that sperm desensitization through GC dephosphorylation is essential for long-range chemotaxis in sea urchins and starfish. In other species ascidian sperm exhibit different signal transduction, in which the sperm responds to the dissociation of the chemoattractant from its receptor [33], whereas in other sea urchins, *Lytechinus pictus* and *Strongylocentrotus purpuratus*, the gradient of chemoattractant evokes [Ca^2+^]_i_ oscillation in the sperm, producing a response involving a series of turns toward the egg [34]. It is possible that desensitization system functions in long-range chemotaxis of these animals, even though the signaling pathways are not similar. Moreover, adjustment of sperm sensitivity would be needed for successful fertilization also in complex animals such as mammals, in which a chemoattractant can be more than one. 

The difference in the desensitization procedure between the sea urchin and starfish appeared in the arrival ratio to the egg and robustness against disturbances. The starfish model showed chemotactic behavior over a longer range and had wider robustness, whereas sea urchins exhibited more efficient chemotaxis over a shorter range than 1200 μm. Although the circulation radius was slightly related to the arrival ratio (Appendix A), sperm of sea urchins with continuous desensitization and starfish with discrete desensitization appeared specific to chemotaxis over the shorter range and the longer range, respectively. Additionally, discrete desensitization appeared to generate wider robustness in starfish, suggesting that starfish sperm does not require precise uniformity through its formation. These differences could be related to reproductive strategies, such as the quality of gametogenesis, spawning distance, and the number of gametes. Molecular recognition for the acrosome reaction, the penetration mechanism of the egg coat, and other fertilization processes are also differentiated between sea urchins and other echinoderms [35]. Therefore, subsequent investigations of starfish were expected to provide general features of echinoderms.

The balance of sperm affinity and sensitivity for chemoattractants appear important, although a certain width of robustness was shown in simulations (Figure 4A–F). Because both (Chemoattractant)_s_ and sperm sensitivity are the feature values of the cell, egg, and sperm, respectively, they could have a certain amount of variation between cells because of the difference in intracellular metabolism. Moreover, a group of eggs would form a much higher local (Chemoattractant) than (Chemoattractant)_s_. Thus, there should be some robustness within these parameters, and as a result our models have functioned, although we have roughly approximated parameters.

In some instances, fluctuations in the relative position between the sperm and the egg resulted in an enhanced rate of successful chemotaxis (Figure 4G,H). Noise and switching of the gain of sperm have also been the focus of mathematical research [36]. Therefore, we suggested that subsequent investigations should examine the correlation between successful fertilization and tidal or thermodynamic positional fluctuations to clarify whether sperm capitalize on these fluctuations. Additionally, the algorithm used in the model can be applied to the optimization algorithms as a type of random walk Monte Carlo method; thus, our model could be the subject for the study of mathematical models.

In a natural environment, such as in the sea, sperm swim in 3D space rather than on a 2D plane [37,38]. Moreover, the sperm curvature radius changes continuously with the time derivative of [Ca^2+^]_i_ [39]. None of these complex phenomena have been included in this study; however, our model advances in the physical characterization of sperm [40,41]. The concept of desensitization theory will expand the understanding of fertilization, followed by medical applications such as infertility treatment in the future.

In this study, we reconstructed sperm chemotaxis in silico, and the models showed chemotactic behavior in the simulation. Our systems biological approach will be useful for the research of sperm chemotaxis in other species to validate whether pathways, parameters, and algorism are sufficient. Moreover, it is applicable for research on the evolution of reproductive strategies by comparing parameters and sperm behaviors in relative species.

## 4. Materials and Methods

### Simulation Experiments

The simulation program was written in the C language with the Mersenne Twister pseudo-random number. The basic source codes of the sea urchin and starfish are shown in Appendix A, respectively. The parameters modified are mentioned in Figure 4A–H.

As mentioned in the Results section, modelling is based on previous experimental findings. Sperm trajectories were simplified from graphs in Figures 2 and 8 of Böhmer et al., 2005 [15]. Sensitivities of sperm were constructed from numerical data of Bönigk et al., 2009 [18], Kaupp et al., 2003 [16], and Nishigaki et al., 2000 [11]. Characteristics of GC-chemoattractant binding were determined from values in result of Pichlo et al., 2014 [21], and Nishigaki et al., 2000 [11].

## Figures and Tables

**Figure 1 ijms-22-09104-f001:**
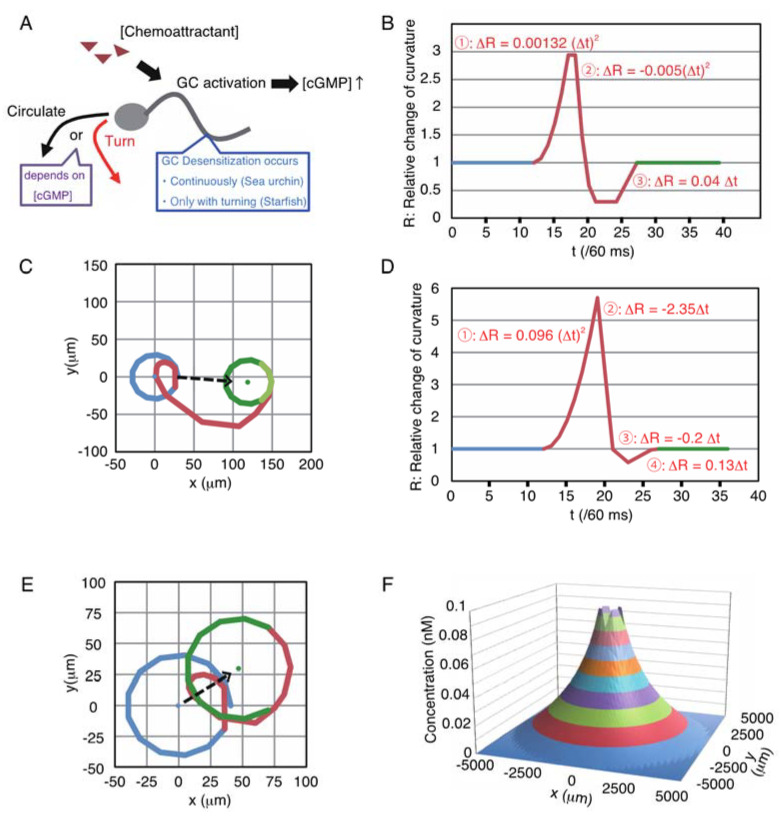
Design of sperm models. (**A**) Conceptual diagram of modeled sperm. Sperm movement was chosen between the circular motion and turn, depending on (Chemoattractant) and (GC). B and D: Sperm curvatures of sea urchin (**B**) and starfish (**D**) in turn mode. The *y*-axis indicates the curvature in relative change to the circular-motion mode. The *x*-axis is a time series in 60 ms frames. C and E: Sperm movement pattern of turn mode in the sea urchin model (**C**) and starfish model (**E**). The colors of the tracks correspond to the colors of lines B and D. (**F**) An example of the shape of the (Chemoattractant) gradient defined within a radius of 5000 μm. The *z*-axis indicates (Chemoattractant) at each point calculated in the 2D surface represented as an x-y plane, where the egg is located at the origin with (Chemoattractant)s = 0.1 nM.

**Figure 2 ijms-22-09104-f002:**
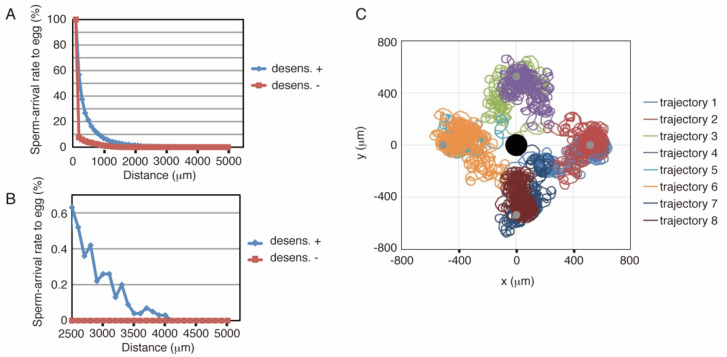
Results of simulations with the sea urchin model. (**A**) Summary of simulation results. The *x*-axis indicates the distance between the sperm start-point and the center of the egg, and egg-arrival rates of sperm from these points are shown as the longitudinal axis (*n* = 10,000). The blue symbols represent the results of the desensitization model, and the red symbols represent the results of a model in which the desensitization function is frozen. The length of the simulation was 30 min. (**B**) Magnification of a part of A. (**C**) Representative trajectories in the simulation. The black circle at the center indicates the egg, and the sperm started swimming from the four gray circles. Trajectories 1 and 2, 3 and 4, 5 and 6, and 7 and 8 start from the same points. Trajectories 2, 4, 6, and 8 show successful examples, whereas the others represent failures.

**Figure 3 ijms-22-09104-f003:**
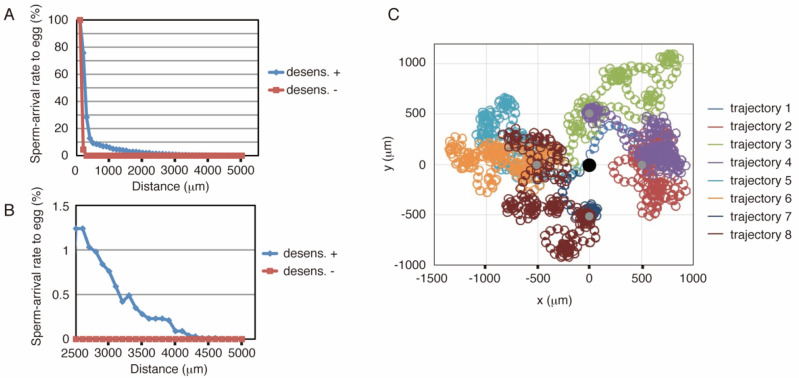
Result of simulations with starfish model. (**A**,**B**) Summary of the simulation results with the starfish model. The axes and colors are the same as in Figure 2A,B. (**C**) Representative trajectories in the simulation using the starfish model. The black circle indicates the egg, and the sperm started from the four gray circles. Trajectories 1 and 2, 3 and 4, 5 and 6, and 7 and 8 start from the same points. Trajectories 2, 4, 6, and 8 show successful examples, whereas the others represent failures.

**Figure 4 ijms-22-09104-f004:**
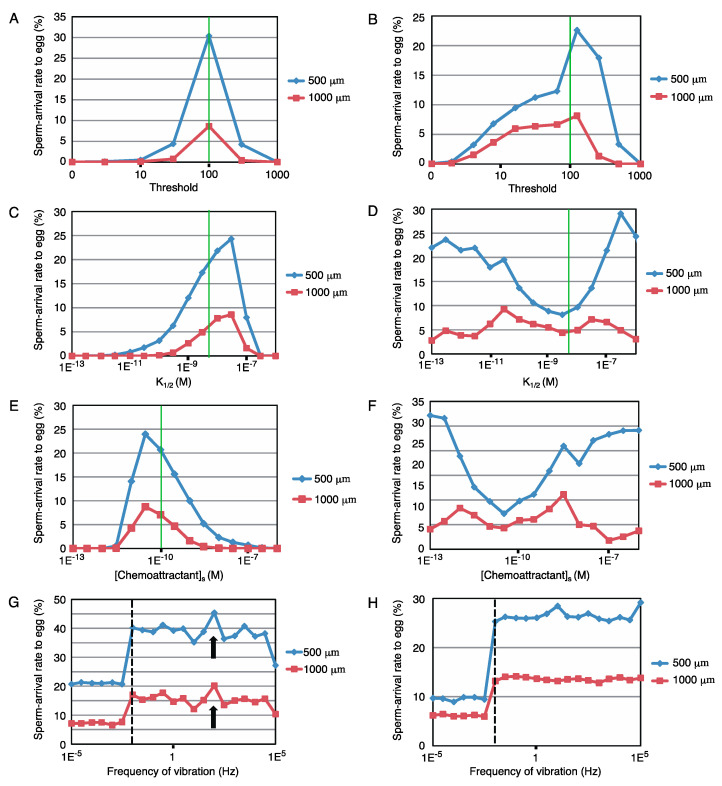
Further analysis of models. (**A**–**F**) Summarized graphs of simulations with modification in parameters and the threshold for [cGMP]_i_, K_1/2_, and [Chemoattractant]_s_. The *y*-axis shows the egg-arrival rates of sperm (*n* = 10,000), with the modification in each parameter shown as the abscissa axis. (**A**,**C**,**E**) correspond to the sea urchin models. (**B**,**D**,**F**) represent the starfish models. Green lines indicate the original values of the parameters. (**G**,**H**) Effects of physical disturbances in the sea urchin model (**G**) and starfish model (**H**). The *y*-axis shows the egg-arrival rates of sperm (*n* = 10,000), with the frequency of sine-wave oscillation shown as the *x*-axis. The broken line represents 0.01 Hz, and the arrowhead in G indicates a local maximum at 100 Hz.

## Data Availability

Data is contained within the article or supplementary material.

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
