# Peer review of "In Silico Reconstruction of Sperm Chemotaxis"

_ijms, 2021, doi:10.3390/ijms22179104_

Round 1

Reviewer 1 Report

The movement of sperm in the direction of the egg is essential for fertilization in the natural cycle. Nevertheless, much more often in papers, researchers pay attention to motor activity, and not to its direction. In the presented article, the authors touch upon an urgent problem and use computer modeling to describe the directional movement of spermatozoa. Overall, the article, and especially the supplementary material, makes a great impression. However, there are some notes:

The main thing is that the authors throughout the text, in particular, in lines 211-213, talk about the applicability of their model to various systems, even if the signaling pathways for activating the directed movement of spermatozoa are different. Maybe it is so. But the model is based on the idea of ​​an initially circular stochastic motion, which further, remaining circular, acquires a direction. In other words, the center of the circumscribed circles is shifted towards a higher concentration of chemoattractant. And this significantly imposes restrictions on the applicability of the model. Such motor activity is more typical for lower animals, such as echinoderms, but for higher animals it is more a variant of pathology than a norm. And, accordingly, it questions the applicability of similar chemotaxis modeling for other species. I urge the authors to be careful about making such statements in the text and to add a paragraph describing the applicability and limitations of their approach.

The lack of formulas describing motor activity significantly complicates the perception. The ratios presented in the text do not allow tracing the initial data and boundary conditions. Authors should introduce basic relationships in section 2.1.

In addition, in the same section, a large number of appeals to the data obtained in previous studies, without at least a brief description of them, does not allow us to evaluate the presented approach. Perhaps authors should make a short summary of the previous data on which their model is based.

In lines 221-222, the comparison with the olfactory system is very pertinent. However, further, in lines 223 - 224, the comparison looks like a part of fiction, not a scientific article.

Author Response

Reviewer #1

[Comment 1]

The main thing is that the authors throughout the text, in particular, in lines 211-213, talk about the applicability of their model to various systems, even if the signaling pathways for activating the directed movement of spermatozoa are different. Maybe it is so. But the model is based on the idea of an initially circular stochastic motion, which further, remaining circular, acquires a direction. In other words, the center of the circumscribed circles is shifted towards a higher concentration of chemoattractant. And this significantly imposes restrictions on the applicability of the model. Such motor activity is more typical for lower animals, such as echinoderms, but for higher animals it is more a variant of pathology than a norm. And, accordingly, it questions the applicability of similar chemotaxis modeling for other species. I urge the authors to be careful about making such statements in the text and to add a paragraph describing the applicability and limitations of their approach.

  • [Answer]

We agree with the comment. As Reviewer #1 pointed out, there’s limitations on the applicability of the model. Therefore, we’d like to revise corresponding paragraph as follows.

Revised paragraph

The results showed that simulations without desensitization lost the chemotactic function (Figs. 2A-B and Figs. 3A-B), suggesting that sperm desensitization through GC dephosphorylation is essential for long-range chemotaxis in sea urchins and starfish. In other species, ascidian sperm exhibit different signal transduction, in which the sperm responds to the dissociation of the chemoattractant from its receptor [32], whereas in other sea urchins, Lytechinus pictus and Strongylocentrotus purpuratus, the gradient of chemoattractant evokes [Ca2+]i oscillation in the sperm, producing a response involving a series of turns toward the egg [33]. It is possible that desensitization system functions in long-range chemotaxis of these animals, even though the signaling pathways are not similar. Moreover, adjustment of sperm sensitivity would be needed for successful fertilization also in complex animals like mammals, in which chemoattractant can be more than one.

[Comment 2]

The lack of formulas describing motor activity significantly complicates the perception. The ratios presented in the text do not allow tracing the initial data and boundary conditions. Authors should introduce basic relationships in section 2.1.

In addition, in the same section, a large number of appeals to the data obtained in previous studies, without at least a brief description of them, does not allow us to evaluate the presented approach. Perhaps authors should make a short summary of the previous data on which their model is based.

  • [Answer]

We agree with the comment. We added the following paragraph in Materials and Methods.

Added paragraph in Materials and Methods

As mentions in Result, modelling is based on previous experimental findings. Sperm trajectories were simplified from graphs in Fig. 2 and 8 of Böhmer et al, 2005 [15]. Sensitivities of sperm were constructed from numerical datas of Bönigk et al, 2009 [18], Kaupp et al, 2003 [16], and Nishigaki et al, 2000 [11]. Characteristis of GC-chemoattractant binding were determined from values in result of Pichlo et al, 2014 [21], and Nishigaki et al, 2000 [11].

[Comment 3]

In lines 221-222, the comparison with the olfactory system is very pertinent. However, further, in lines 223 - 224, the comparison looks like a part of fiction, not a scientific article.

  • [Answer]

We agree with the comment. We deleted the lines 223 – 224.

Reviewer 2 Report

In the present study the authors propose an algorithm for sperm chemotaxis in sea urchin. Although the research conducted is interesting, the manuscript needs deep revision before its acceptance for publication.

Discussion

  • Authors must revise this section and highlight the biological significance of the study and potential applications.

Material and Methods

  • The information given is too scarce. Considering that the modelling is based on previous experimental findings, authors must cite these studies and describe the experimental data they used to develop the algorithm.

Conclusions

  • Authors must include a Conclusions section highlighting the biological relevance of the proposed model.

Authors must also review the manuscript and correct spelling and grammatical errors.

Author Response

Reviewer #2

[Comment 1]

Discussion

Authors must revise this section and highlight the biological significance of the study and potential applications.

  • [Answer]

We fundamentally changed the structure of “Discussion section” to highlight the significance of our works.

At first, we explain about the most important point, concept of desensitization theory. Then, we described about potential applications in biology and mathematics respectively at second and forth paragraphs. Also, we mentioned medical applications at the end of section.

[Comment 2]

Material and Methods

The information given is too scarce. Considering that the modelling is based on previous experimental findings, authors must cite these studies and describe the experimental data they used to develop the algorithm.

  • [Answer]

We added paragraphs explaining the relation of parameters in the model and previous studies as follows.

Added paragraph

As mentions in Result, modelling is based on previous experimental findings. Sperm trajectories were simplified from graphs in Fig. 2 and 8 of Böhmer et al, 2005. Sensitivities of sperm were constructed from numerical datas of Bönigk et al, 2009, Kaupp et al, 2003, and Nishigaki et al, 2000. Characteristis of GC-chemoattractant binding were determined from values in result of Pichlo et al, 2014, and Nishigaki et al, 2000.

[Comment 3]

Authors must include a Conclusions section highlighting the biological relevance of the proposed model.

  • [Answer]

We added “Conclusion section” as reviewer#2 mentioned, which highlights significance of our approach as follows.

Conclusions

In this study, we reconstructed sperm chemotaxis in silico, and the models showed chemotactic behavior in the simulation. Our systems biological approach will be useful for the research of sperm chemotaxis in other species to validate whether pathways, parameters, and algorism are sufficient. Moreover, it is applicable for research on the evolution of reproductive strategies by comparing parameters and sperm behaviors in relative species.

Round 2

Reviewer 2 Report

Authors have improved the quality of the manuscript according to reviewers' comments. Nevertheless, there ara some minor mistakes that authors must correct before its acceptance for publication:

  1. Line 31. Please, write Homo sapiens in italics.
  2. Lines 80-81. Please, write Ar. punctulate in italics
  3. Line 94. Please, eliminate bold format.
  4. Lines 99-100. "The trajectories are nearly identical 99
    to those reported by Böhmer et al. [25]." This sentence must be placed in the Results or Discussion section instead of the figure legend.
  5. Results section. Please, use the past tense in the whole section.

Author Response

Point-by-point response to the reviewer’s comments

Dear reviewers

We here described explanations corresponding to each comment by reviewers, which includes how we revised our manuscript by advises of reviewers.

Masahiro Naruse

Midori Matsumoto

Comments and Suggestions from Reviewer

  1. Line 31. Please, write Homo sapiens in italics.
  2. Lines 80-81. Please, write Ar. punctulate in italics
  3. Line 94. Please, eliminate bold format.
  4. Lines 99-100. "The trajectories are nearly identical 99
    to those reported by Böhmer et al. [25]." This sentence must be placed in the Results or Discussion section instead of the figure legend.
  5. Results section. Please, use the past tense in the whole section.

  • [Answer]

Thank you for the accurate comments.

    1, 2.       changed to italics.

  1. eliminated bold format.
  2. We transferred “The trajectories were nearly identical to those reported by Böhmer et al. [25].” from figure legend to Results.

    5.  We changed to the past tense in the whole section.

.
